# A Variable-Volume Heart Model for Galvanic Coupling-Based Conductive Intracardiac Communication

**DOI:** 10.3390/s22124455

**Published:** 2022-06-12

**Authors:** Yiming Liu, Yueming Gao, Liting Chen, Tao Liu, Jiejie Yang, Siohang Pun, Mangi Vai, Min Du

**Affiliations:** 1College of Physics and Information Engineering, Fuzhou University, Fuzhou 350108, China; zsemju@foxmail.com (Y.L.); clt_9807@163.com (L.C.); litoyu@outlook.com (T.L.); chy@fzu.edu.cn (J.Y.); dm_dj90@163.com (M.D.); 2State Key Laboratory of Analog and Mixed-Signal VLSL, University of Macau, Macao, China; lodgepun@umac.mo (S.P.); fstmiv@umac.mo (M.V.)

**Keywords:** leadless pacemaker, conductive intracardiac communication, galvanic coupling, equivalent heart model, volume of chamber, circuit-coupled electrical field model

## Abstract

Conductive intracardiac communication (CIC) has become one of the most promising technologies in multisite leadless pacemakers for cardiac resynchronization therapy. Existing studies have shown that cardiac pulsation has a significant impact on the attenuation of intracardiac communication channels. In this study, a novel variable-volume circuit-coupled electrical field heart model, which contains blood and myocardium, is proposed to verify the phenomenon. The influence of measurements was combined with the model as the equivalent circuit. Dynamic intracardiac channel characteristics were obtained by simulating models with varying volumes of the four chambers according to the actual cardiac cycle. Subsequently, in vitro experiments were carried out to verify the model’s correctness. Among the dependences of intracardiac communication channels, the distance between pacemakers exerted the most substantial influence on attenuation. In the simulation and measurement, the relationship between channel attenuation and pulsation was found through the variable-volume heart model and a porcine heart. The CIC channel attenuation had a variation of less than 3 dB.

## 1. Introduction

The traditional pacemaker has been evolving for decades and is currently a mature and effective therapeutic for certain cardiac diseases. However, clinical reports and long-term studies show that the lead wires are the primary causes of complications, contributing to more than 10% of early complications within the first two months after implantation [1]. Therefore, leadless pacemakers (LCPs) have emerged and are fabricated with state-of-the-art electronic technology to eliminate the leads of a traditional pacemaker. Although they have the advantages of small size and no lead wires, leadless pacemakers can only perform single ventricular pacing (right ventricle pacing) to be precise [2], failing to address cardiac resynchronization therapy (CRT). For patients with moderate-to-severe heart failure, CRT with multiple pacemakers is the preferable therapeutic [3]. In CRT, multiple pacing leads are implanted into the right atrium, right ventricle, and left ventricle to regulate the contraction rhythm of each chamber of the heart and realize the regular beating of the heart. To achieve CRT and leverage the advantages of the LCP, multiple LCPs can be implanted into cardiac chambers to implement a multisite leadless pacemaker system for CRT. However, LCPs are unsynchronous and hard to coordinate unless equipped with efficient communication methods. Thus, highly reliable and low-power-consumption wireless communication is critical in multisite LCP systems to realize CRT.

With the dielectric properties of heart tissues, galvanic coupling (GC) has been proposed to realize conductive intracardiac communication (CIC) for multisite LCP systems. In GC, a transmitted signal is applied to the human body through two transmitting electrodes and received by pair of electrodes in a remote site [4] within the human body. Compared with wired and classic wireless communication technology, GC-based CIC (GCCIC) does not need an antenna, is simple and reliable, and has robust anti-interference, low transmission power consumption, and slight external radiation [5]. The power consumption of GC is only 3.7 mW, while the data rate reaches 6 Mbps [6]. Previous research has shown that GCCIC is suitable for intracardiac communication, with myocardium and blood as conductors. From the literature, blood and myocardium are the principal constituents of the human heart and have low impedance. This property can boost GCCIC, and in contrast to electromagnetic-based communication technologies, low impedance weakens their efficiencies. Lukas Bereuter et al. [7,8] were the first to propose GCCIC technology for intracardiac communication. They performed in vitro and in vivo measurements on a porcine heart and used the finite element model (FEM) simulation in the frequency range of 1 kHz to 1 MHz to study the signal propagation and impedance behavior of blood and myocardial tissue. In addition, they constructed a prototype pacemaker and performed CRT pacing three times. Mirko Maldari [9] designed an accurate human heart model to calculate the path loss of intracardiac communication channels and verified its performance using experimental measurements. These pilot works provide a sound basis for multisite LCP systems. However, their proposed model is complicated and has low computational efficiency. Furthermore, the existing models only consider intracardiac communication channels under static conditions and generally neglect the influence of cardiac beats.

The cardiac beat causes geometrical shape and electrical characteristic variations in intracardiac channels. It can inevitably affect the communication quality of GCCIC, which has been confirmed in existing studies [7]. Without a valid model incorporating dynamic variations in the heart, accurate and realistic channel parameters cannot be obtained to design a reliable CIC for multisite LCP systems. We performed a preliminary exploration using the equivalent heart model [10]. In this study, we propose a novel variable-volume finite element equivalent heart model to analyze the dynamic channel characteristics within a cardiac cycle. We aimed to determine the dependence of different factors on attenuation in intracardiac communication channels by performing model simulations and experiments on a porcine heart. To facilitate the study of CRT closer to current clinical applications, this study explored two intracardiac communication channels: (1) between the right and left ventricles (RV-LV); (2) between the right ventricle and right atrium (RV-RA).

The remaining parts of the paper proceed as follows: In Section 2, a FEM is presented, and in vitro experiments on the intracardiac communication channel are described. Then, all the simulation and measurement results are reported in Section 3. Finally, the conclusions of this article are drawn in Section 4.

## 2. Materials and Methods

The main purpose of this study is to analyze the channel characteristics of GCCIC in intracardiac channels. A FEM is proposed for circuit-coupled electrical field analysis. Based on the traditional electromagnetic field simulation, the model has the advantage of variable volume and thickness of myocardial tissue. In addition, the electric field characteristics in the model vary, such as the signal frequency, input and output impedance, and so on. Dynamic intracardiac channel characteristics are obtained by simulating models with the volumes of the four heart chambers varied according to the actual cardiac cycle. For model validation, experiments were carried out on a porcine heart.

### 2.1. Circuit-Coupled FEM Model for Conductive Intracardiac Communication

All simulations were carried out using COMSOL Multiphysics, version 5.4. The FEM with GCCIC was developed with the AC/DC branch of COMSOL Multiphysics. The dependence of attenuation on the following factors was explored by conducting simulations based on the FEM model: input impedance; thickness of myocardial tissue (outer myocardium and interventricular septum); and distance between transmitter and receiver.

The human heart is primarily composed of the myocardium, blood, epicardium, endocardium, and other tissues [11]. As the GCCIC signal mainly flows through the blood and myocardial tissues, the heart was simplified and represented by an equivalent model consisting of blood and myocardial tissue (Figure 1). The simplified heart model consisted of a hemisphere structure with a radius of 47.3 mm and a semi-ellipsoid with a short axis of 47.3 mm and a long axis of 63.5 mm. There were four chambers in the model, representing four cardiac chambers, i.e., left atrium (LA), left ventricle (LV), right atrium (RA), and right ventricle (RV). Table 1 shows the volume of each chamber [12,13,14]. In the cardiac cycle, only the ventricular volume is changed, and the atrial volume remains unchanged.

Cylindrical capsules with shapes similar to the commercial LCP [2,15] were placed inside the heart model, as illustrated in Figure 1. The LCP (cylindrical capsule) had a length of 23 mm and a radius of 4.5 mm. The upper and lower bottom surfaces were configured as conductive electrode surfaces, and the rest were set as insulation characteristics. In this work, three pacemakers were placed in RV near the apex, RA near the right atrial appendage, and LV near the apex, mimicking the CRT pacing in clinical applications.

The model’s blood volume and myocardial thickness were designed to be varied to examine the influence of different attenuation factors of intracardiac channels. Simulating different volumes of the four chambers could also reveal attenuation channel changes during the cardiac cycle. The thickness of the outer myocardium was set to 3 mm, while the thickness of the interventricular septum was set to 10 mm. In the study of myocardial tissue thickness, the thickness of the outer myocardium increased from 2 mm to 10 mm, and the thickness of the interventricular septum increased from 10 mm to 20 mm. The step sizes of the thickness of the outer myocardium and the interventricular septum were both 2 mm.

In order to explore the effect of channel length on intracardiac communication, the distance between the transmitter and receiver (LCPs) was gradually increased. The LCP in RV was set as the transmitter, and the LCPs located at RA and LV were set as the receivers. Thus, two main intracardiac communication channels for GCCIC were constructed for the multisite LCP system and realized the standard CRT pacing. The LCPs at RV and LV formed the RV-LV channel, and the LCPs at RV and RA formed the RV-RA channel. To study the attenuation by the separation distance between the transmitter and receiver, we used the RV-RA channel, and the separation distance varied from 3 to 9 mm.

The dielectric parameters of the myocardium and blood, including relative permittivity and conductivity, were introduced into the model. The dielectric properties of tissues at various frequencies were calculated using the Cole–Cole model according to Gabriel et al. [16,17].

The finite element mesh was generated with the Free Tetrahedral Meshing tool in COMSOL Multiphysics. The maximum and minimum sizes were 11.1 mm and 2.0 mm, respectively, and the number of degrees of freedom was 45598. The Multifrontal Massively Parallel sparse direct Solver (MUMPS) was used to calculate the partial differential equations.

The frequency range of the transmitted signal was set from 10 kHz to 10 MHz, which satisfied the quasi-static conditions so that the inductive and wave propagation effects were negligible [18,19]. Therefore, Maxwell’s equations could be simplified for the quasi-static electric field:(1)∇[(σ+jωε)∇V=0

The circuit model was set up in the COMSOL interface, and the circuit was combined with the FEM of the equivalent heart model, as shown in Figure 2.

The equivalent circuit models of the transmitter (left) and receiver (right), as depicted in Figure 2, were incorporated into the simulator to represent the electrical properties of the transmitter and receiver. For the transmitter, an AC power source *V_T_* with internal resistance *r* (50 Ω) was connected in series with the transmission electrode pair. At the receiver side, an input impedance *R_i_* was connected in series with the receiving electrode pair to represent the input impedance of the devices. The receiving voltage *V_R_* was obtained through the electrodes at the input impedance *R_i_*. In the simulation, we selected six resistance values, 50, 100, 1 k, 10 k, 100 k, and 10 MΩ, to analyze the influence of input resistance in the intracardiac channel. The purpose was to clarify whether the increase in input impedance would reduce the attenuation of the intracardiac channel so as to provide a basis for the design of the pacemaker transceiver. The current field interface and the circuit interface were connected via electrodes to realize the field circuit combination. Attenuation of GCCIC was calculated:(2)$$A=20\cdot \mathrm{lg}\left(\frac{V_{R}}{V_{T}}\right)$$

### 2.2. Variable Volume with the Cardiac Cycle

During each cardiac cycle, the volume of each chamber periodically changes because of the myocardial contraction. Based on the max–min volume of each chamber, the cardiac cycle was divided into three phases by three distinct time points: end-systolic (ES), mid-diastolic (MD), and end-diastolic (ED) [11]. Table 1 and Figure 3 show the volume of each chamber at different time nodes. ES is the time instant when the ventricle contracts to the minimum, while ED corresponds to when the ventricular diastole reaches its maximum value. MD is the time point of a sudden change in ventricular diastolic pressure. From ED to ES, the ventricle pushes most of the blood into the artery, and the ventricular volume decreases from the maximum to the minimum. From ES to MD, ventricular dilation produces suction, and approximately three-quarters of the circulating blood flows from the atrium into the ventricle. When the pressure of the atrium and ventricle is equal (MD), the filling of the ventricle temporarily stops. Then, from MD to ED, atrial systole increases the pressure gradient from the atrium to the ventricle, which causes the remaining quarter of the blood to flow into the ventricle. The interval between the three principal time points is reduced by selecting a time node between each two main time nodes (TN1, TN2, and TN3).

Due to the variable volume, equivalent heart models at each time node were obtained to simulate the beating of the heart. Firstly, the interventricular septal thickness (10 mm) was consistent with the previous model. Then, according to the volume of four cardiac chambers at different time nodes, the radius of the hemisphere and semi-ellipsoid was adjusted to change the volume of chambers in FEM. Finally, a 3 mm thick outer layer of the outer myocardium was formed outside the cardiac chamber. The attenuations of the intracardiac channel at different time nodes were analyzed by simulating the models at each time node and arranged in time order. During the change in the volume of each chamber, the thickness of the myocardium remained unchanged. The pacemakers were always fixed at the corresponding pacing point, maintaining a consistent relative spatial angle, and the dielectric properties of the myocardium were set to isotropic.

### 2.3. In Vitro Experiment

To verify the channel characteristics of GCCIC from FEM, an in vitro measurement (Figure 4) was designed to measure the characteristics of intracardiac communication channels. A high-precision vector network analyzer (VNA E5061B 3L5) with a frequency range of 10 kHz to 10 MHz was adopted to measure the attenuation of the channel. The GCCIC ground was decoupled from the equipment ground using a differential probe (1141A Differential Probe and 1142A Probe Conduct) to detect the received signal [20].

In the in vitro experiment, three incised porcine hearts, similar to the human heart in morphology, tissue structure, and dielectric properties, with weights of 0.49, 0.44, and 0.40 kg were used for the in vitro experiment.

The electrodes of the LCP were inserted into the heart myocardium, and their radius was 4.5 mm, as in the FEM. The porcine hearts were cut longitudinally, and the electrodes were inserted into the muscle with the conductive electrode facing outward and an ipsilateral electrode spacing of 23 mm. The electrodes were connected to the VNA, forming two communication channels, i.e., RV-LV and RV-RA. In this study, the transmitter impedance of the VNA was 50 Ω, and the receiver impedance of the differential probe was 1 MΩ.

### 2.4. Dynamic In Vitro Experiment

To study the communication properties of GCCIC under the influence of heart beating, a dynamic experimental platform was constructed with a programmable controller (PLC), a driver, a stepping motor, the VNA, and a differential probe (a photo of the experimental setup can be found in Figure 5).

As Figure 5 shows, the controller and the stepping motor were driven by the driver. The stepping motor expanded according to the speed and distance set by the controller. In this study, the stepper motor was connected with the porcine hearts by an S-shaped part, and the periodic movement of the stepper motor drove the heart to relax and contract to simulate the beating process of the heart. Due to the limitation of experimental conditions, external connections were not used for blood pumping in and out in in vitro experiments. The simulation of the experimental central dynamic cycle was only realized on the myocardium. During the process, the heart was filled with blood to eliminate the influence of air on the experimental results.

Table 1 and Figure 3 show the volume change in each chamber at different time nodes. In a complete cardiac cycle, the stepper motor advanced a total distance of 40 mm, and the systolic and diastolic processes were commensurate with 20 mm. For ES to MD, the ventricle expanded and allowed three-quarters of the circulating blood to flow in from the atrium, corresponding to three-quarters of the diastole, which was 15 mm. For MD to ED, the rest of the blood in the atrium flowed into the ventricle, and the ventricles dilated to the maximum, corresponding to a quarter of the diastole. Consequently, the stepper motor extended 5 mm more. For ED to ES, the ventricle contracted, and thus, the stepper motor was shortened to 20 mm. TN 1, TN 2, and TN 3 are the midpoints of the three periods. The order of the process was ES, TN 1, MD, TN 2, ED, TN 3, and ES, which simulated a normal cardiac cycle.

## 3. Results

### 3.1. Simulation Results

#### 3.1.1. Input Impedance

This section analyzes the input impedance of RV-RA and RV-LV channels. Figure 6 shows that the simulated attenuation of intracardiac communication changes with different input impedances (*R_i_* in Figure 2).

Among the values of input impedance, 50 Ω is the generic measurement equipment value [20]. Figure 6 reveals that the attenuation trend of the two channels is similar. When the input impedance is less than 10 kΩ, the attenuation of the channel decreases with the increase in the input impedance. However, when the input impedance is greater than 10 kΩ, the channel attenuation at each frequency no longer changes significantly. The variation range of attenuation is not more than 2%, and the attenuation curves almost coincide. As the input impedance increases above 100 kΩ, the attenuation can be deduced to have no significant changes. This deduction is verified by increasing the input impedance to 10 MΩ, and the channel attenuation curve remains the same as at 10 kΩ. This may be because when the impedance is low, the increase in impedance will increase the potential difference induced on the receiving electrode, so the gain increases with the increase in the impedance. When the impedance is higher than 100 kΩ, the potential difference that can be received on the receiver has reached the maximum value. At this time, increasing the receiver impedance has little effect on the gain. Thus, input impedance has a definite influence on the attenuation of the intracardiac communication channel. The corresponding input impedance should be tailored depending on the actual measurement.

#### 3.1.2. Thickness of Myocardial Tissue

Figure 7 and Figure 8 show the simulated attenuation with different thicknesses of the myocardial tissue. Received voltages were measured separately at the LCPs of RA and LV by using the LCP at RV as the signal source. To assess the effect of the thickness of myocardial tissue in avoiding interference from other factors, the distance between the transmitter and receiver remained constant while increasing the thickness of myocardial tissue.

The results, as shown in Figure 7, indicate that the attenuation curve tends to decrease for the RV-RA channel as the thickness of the outer myocardium increases from 2 mm to 10 mm. For the RV-LV channel, the attenuation increases with the thickness of the outer myocardium, except for when the outer myocardium is 2 mm thick. Interestingly, the attenuations of the two channels show an opposite trend in Figure 7. A possible explanation for this might be that in the RV-RA channel, the signal propagates more along with the blood, while in the RV-LV channel, it propagates more along the outer myocardium. Moreover, the attenuation fluctuation of the intracardiac channel caused by the change in outer myocardial thickness is less than 1 dB in the frequency range of 10 kHz–10 MHz.

Figure 8 shows that changes in the thickness of the interventricular septum also influence the channel attenuations. In contrast to the thickness of the outer myocardium, the effects of the interventricular septum on the channel attenuations in the RV-RA channel are similar to those in the RV-LV channel. The attenuation of both channels decreases inversely as the thickness of the interventricular septum increases. However, closer inspection of the table shows the attenuation fluctuation of the channel because the change in the thickness of the interventricular septum is less than 0.5 dB.

In summary, the effect of myocardial thickness on attenuation is less than 1 dB in the selected frequency band. The data reported here appear to support the assumption that this factor is not important in the study of intracardiac channel attenuation.

#### 3.1.3. Distance between Transmitter and Receiver

The attenuation in the RV-RA channel with different distances between the transmitter and receiver is shown in Figure 9. The attenuation of the intracardiac channel substantially changes and increases gradually with channel length. An increase of 2 cm between the transmitter and receiver results in attenuation increases of approximately 4–9 dB increments. The attenuation of the longest channel is nearly 20 dB higher than that of the shortest channel in the current experimental setting.

### 3.2. Simulation of Dynamic Model

Figure 10 shows the simulated attenuation in different time nodes in the cardiac cycle. The attenuation of the RV-RA channel fluctuates between −30 dB and −35 dB, while the attenuation of the RV-LV channel ranges between −50 dB and −55 dB. As the frequency increases, the fluctuation of channel attenuation at each frequency slightly increases. Figure 11 more intuitively shows the attenuation characteristics of dynamic intracardiac communication channels within a single heartbeat period at 1 MHz frequency. From ES to ED, there is a gradual increase in the attenuation of the intracardiac communication channel, whilst from ES to MD, the channel attenuation significantly decreases abruptly, and from MD to ED, the decrease in channel attenuation is relatively small. In a single cardiac cycle, the attenuation fluctuation (difference between the maximum and minimum attenuations) of the intracardiac communication channel is 2.46 dB (RV-RA channel) and 1.80 dB (RV-LV channel).

### 3.3. In Vitro Measurements

Figure 12 shows the attenuation simulation and measurements with different frequencies. As the signal frequency increases, the attenuation of the two intracardiac communication channels shows a certain change trend, but the change is not obvious. The attenuation in measurements floats around 33 dB (RV-LV channel) and 51 dB (RV-RA channel). The maximum errors in measurement are about 2.94 dB (RV-LV channel) and 4.51 dB (RV-RA channel). As shown in Figure 12, the prediction of intracardiac communication channels by the model established in this paper is in good agreement with the actual measurement results. The single most striking observation to emerge from the data comparison was for the RV-RA channel; the attenuation curve of the simulation and measurement is almost coincident, except when the frequency is near 10 kHz or 10 MHz. In the frequency band from 20 kHz to 9 MHz, the attenuation obtained by the simulation and experiment is similar in value and change trend. The maximum errors between the simulation and measurement are about 1.47 dB (RV-LV channel) and 1.91 dB (RV-RA channel). This inconsistency may be due to the difference in conductivity and dielectric constant between human and porcine hearts. Given that the simulations compared well with the experiments, the FEM in this paper can be considered suitable for the study of intracardiac channels.

Figure 13 indicates the attenuation changes in a cardiac cycle at 1 MHz. The top half of the table shows the changes in the RV-LV channel, whereas the bottom half shows the changes in the RV-RA channel. The channel attenuation in both channels has the same trend, with the minimum at ES and the maximum reached at ED. From ES to ED, the attenuation of the intracardiac communication channel gradually increases; from ED to ES, the channel attenuation decreases. In one cardiac cycle, the attenuation fluctuations of the RV–RA channel are 2.46 dB (simulation) and 2.59 dB (measurement), while those of the RV–LV channel are 1.80 dB (simulation) and 2.20 dB (measurement). The maximum errors between the simulation and measurement results are 0.62 dB (RV–RA channel) and 0.76 dB (RV–LV channel).

A possible explanation for this might be that at ES, the distance between the transmitter and receiver is the shortest, so attenuation reaches the minimum. At ED, the distance is the longest, and attenuation reaches the maximum. Figure 13 also accords with our earlier observations (Figure 12), which shows that the simulation results of the FE model are in good agreement with the measurement results. The simulation results are slightly larger than those of the measurement, but the trends are consistent.

## 4. Conclusions

Leadless pacemakers effectively overcome the problems of traditional pacemakers due to leading wires. However, the application range of leadless pacemakers is small, as they are limited to single-chamber pacing. By implanting multiple pacemakers, multisite leadless pacemaker systems can meet the physiological pacing needs of the heart. As an emerging communication method, GCCIC takes human tissue as the channel, which presents the advantages of low power consumption, strong anti-interference ability, and no requirements for antennas. CIC can thus be a potential candidate for communication among leadless pacemakers.

This study proposes a novel variable-volume circuit-coupled electrical field heart model with GCCIC. Through simulations, the dependences of the intracardiac channel were examined in the frequency band of 10 kHz to 10 MHz. The results show that an input impedance lower than 10 kΩ has a pronounced influence on the attenuation of the intracardiac channel, and the variation is approximately 8 dB. When input impedance is higher than 1 MΩ, its change has no noteworthy effect on attenuation. In addition, the thicknesses of the outer myocardium and interventricular septum have negligible influences on the attenuation of the intracardiac channel, and the fluctuation is lower than 1 dB. With the increase in the channel length, the attenuation increases gradually, and that of the longest channel is nearly 20 dB higher than that of the shortest channel. The most notable finding to emerge from the results of simulations is that the distance between pacemakers is the most influential factor affecting attenuation. The channel is modeled considering the change in volume in the live beating heart, which has not been considered in other references. In one cardiac cycle, the attenuation fluctuations of intracardiac communication channels are less than 3 dB. This verifies that cardiac pulsation can lead to changes in the attenuation of the intracardiac communication channels. It should be pointed out that the dynamic intracardiac communication channel model in this paper was formed by establishing a static model of multiple heartbeat moments and in the order of heartbeat time, which may lead to a deviation between the results and the real situation.

This study focuses on the attenuation of the intracardiac channels with GCCIC. In general, this study strengthens researchers’ understanding of the attenuation, especially for changes in attenuation during the cardiac cycle. The insights gained from this study may be of assistance to the design of the variable-gain received signal strength indication (RSSI) transceiver analog front end in the cardiac pacemaker field. These findings have significant implications for the understanding of how to further reduce cardiac pacemaker power consumption.

## Figures and Tables

**Figure 1 sensors-22-04455-f001:**
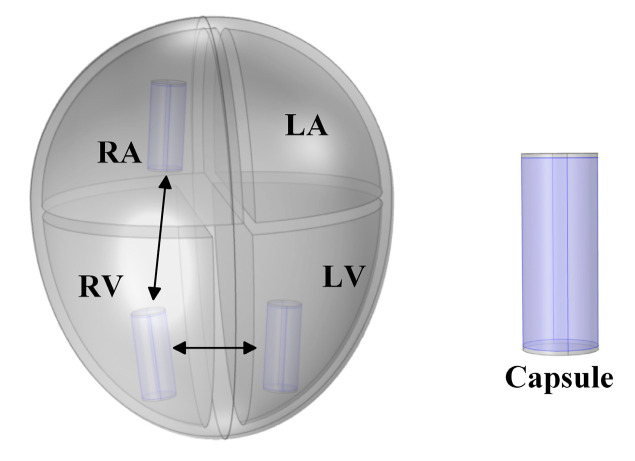
Equivalent heart model divided into four chambers, RA, RV, LA, and LV. Three pacemakers (cylindrical capsules) are positioned inside the chambers.

**Figure 2 sensors-22-04455-f002:**
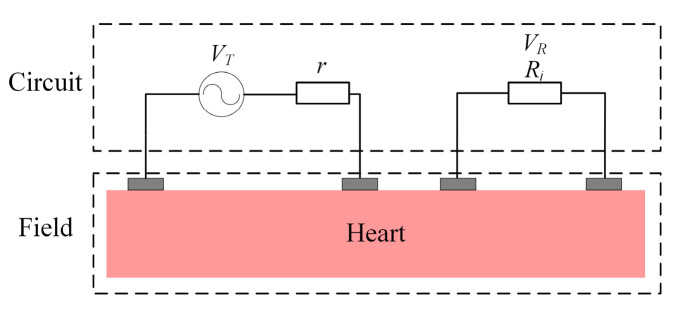
Circuit model of the circuit-coupled FEM analysis.

**Figure 3 sensors-22-04455-f003:**
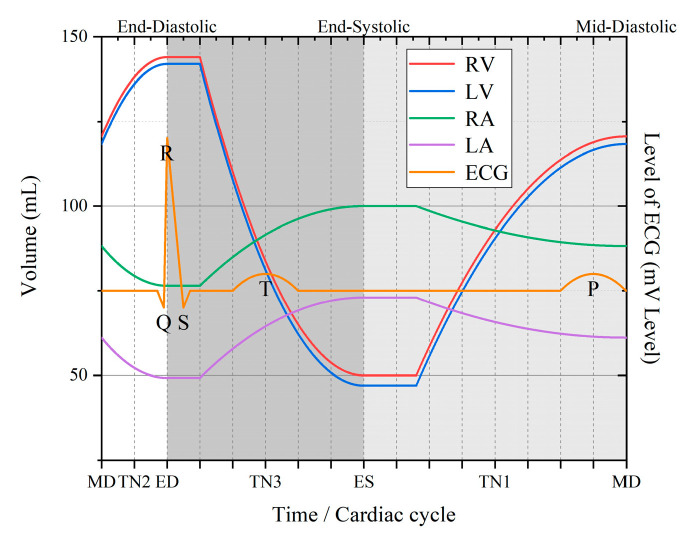
Change curves of ventricular and artery volumes during the cardiac cycle. Diastolic and systolic are distinguished by different gray levels. Different chambers are distinguished by different colors.

**Figure 4 sensors-22-04455-f004:**
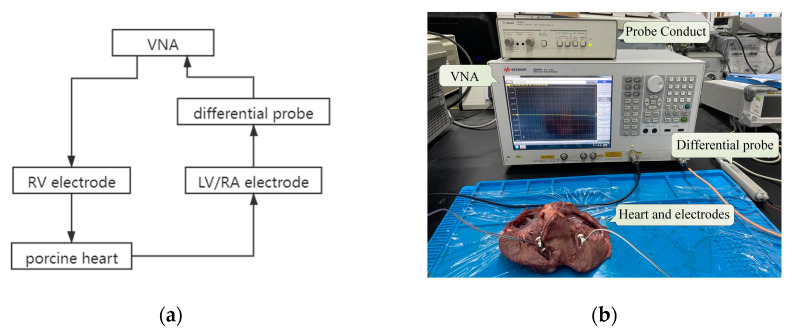
(**a**)The block diagram of in vitro experiment. (**b**) The physical map of the in vitro experiment. Because VNA can only analyze one network at a time, only one channel was studied at every turn.

**Figure 5 sensors-22-04455-f005:**
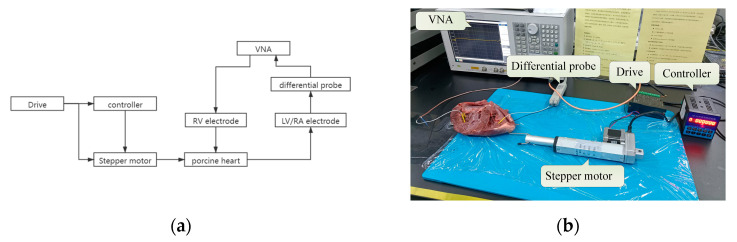
(**a**) The block diagram of dynamic in vitro experiment. (**b**) The physical map of the experimental platform and porcine heart.

**Figure 6 sensors-22-04455-f006:**
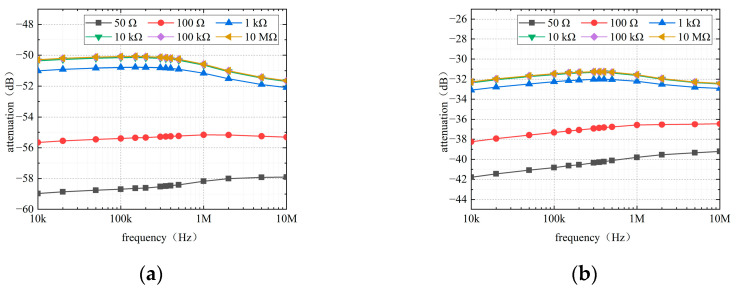
Effect of input impedance on channel attenuation: (**a**) RV-RA channel and (**b**) RV-LV channel.

**Figure 7 sensors-22-04455-f007:**
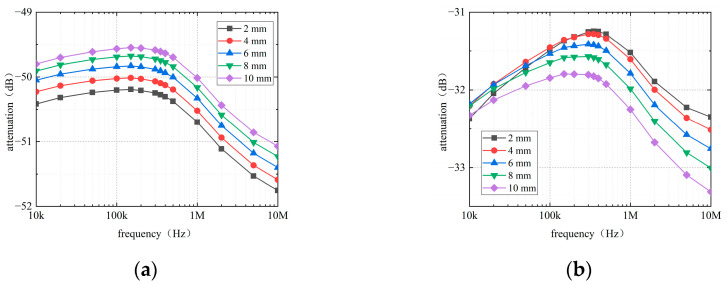
Effect of the thickness of outer myocardium on channel attenuation: (**a**) RV–RA channel, (**b**) RV–LV channel.

**Figure 8 sensors-22-04455-f008:**
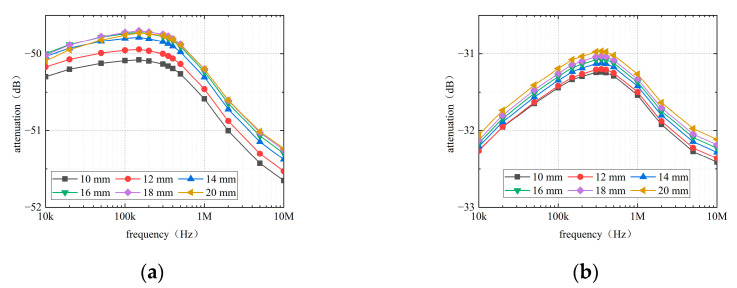
Effect of the interventricular septum on channel attenuation. (**a**) RV-RA channel. (**b**) RV-LV channel.

**Figure 9 sensors-22-04455-f009:**
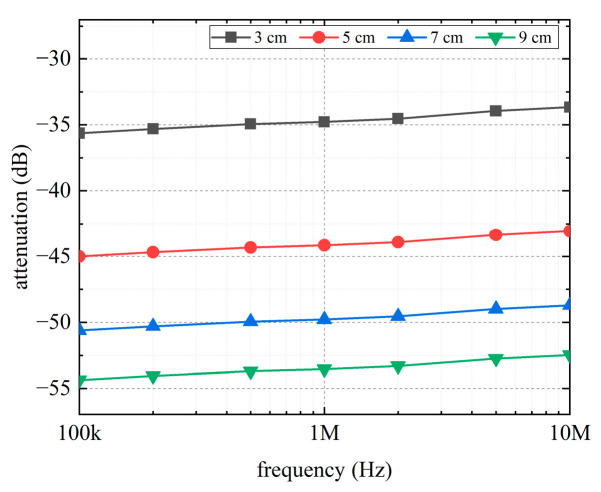
Effect of the distance between transmitter and receiver on the channel attenuation.

**Figure 10 sensors-22-04455-f010:**
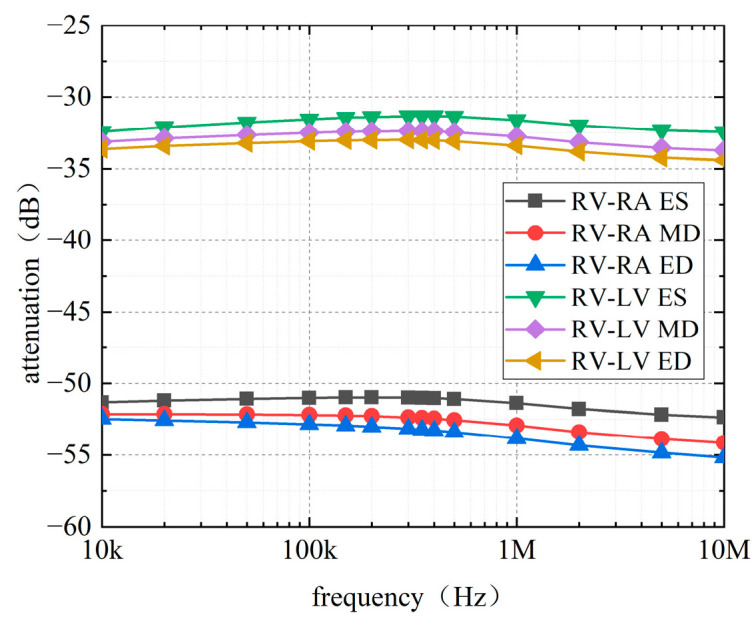
Effect of the distance between transmitter and receiver on the channel attenuation.

**Figure 11 sensors-22-04455-f011:**
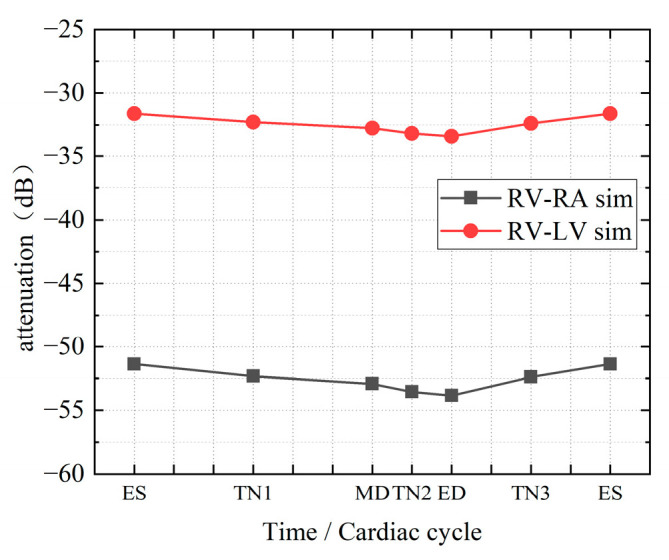
Attenuation time curve of a single cardiac cycle at 1 MHz frequency.

**Figure 12 sensors-22-04455-f012:**
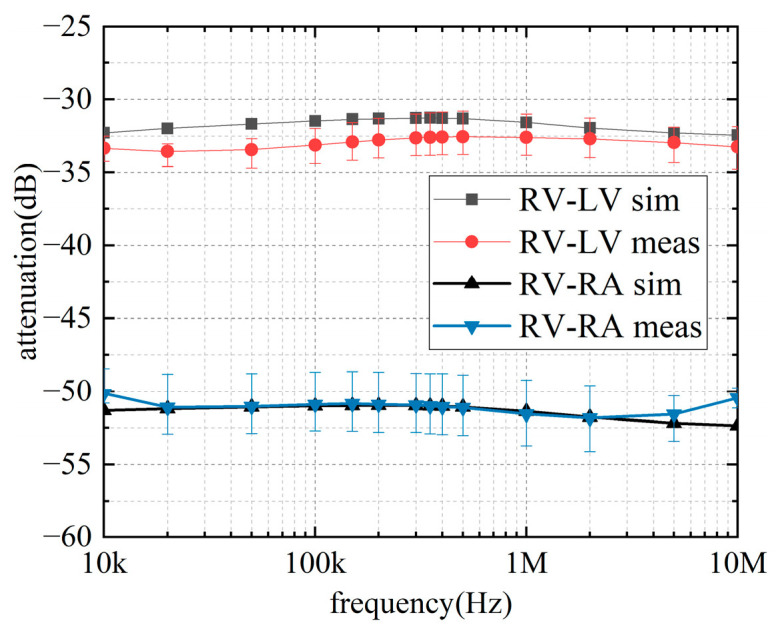
Experimental results of static porcine heart at different frequencies.

**Figure 13 sensors-22-04455-f013:**
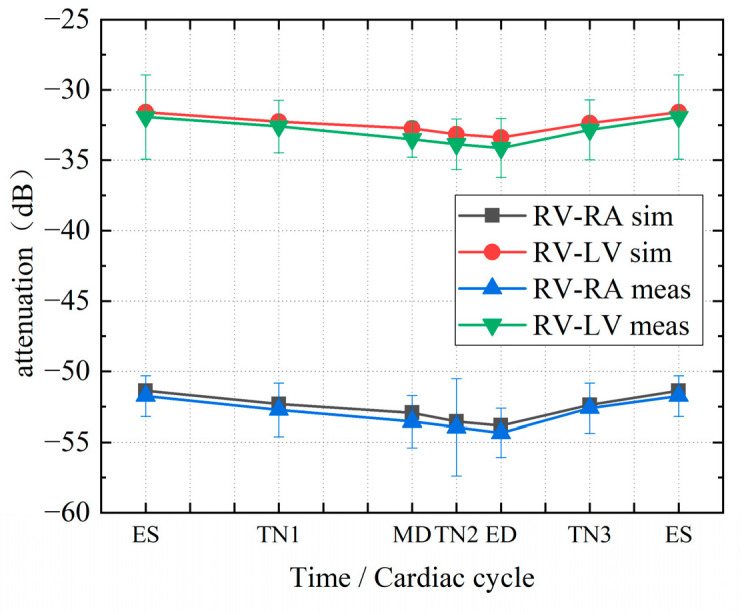
Experimental results of static porcine heart at 1 MHz.

**Table 1 sensors-22-04455-t001:** Volume of Each Chamber.

Chamber	Static Volume (mL)	Dynamic Volume (mL)		
		ES	MD	ED
LA	73 ± 15	73	73	73
LV	95 ± 14	47	118.25	142
RA	100 ± 20	100	100	100
LV	94 ± 15	50	120.5	144

## Data Availability

The data presented in this study are available on request from the corresponding author. The data are not publicly available due to privacy.

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
