# Peer review of "A Variable-Volume Heart Model for Galvanic Coupling-Based Conductive Intracardiac Communication"

_sensors, 2022, doi:10.3390/s22124455_

Round 1

Reviewer 1 Report

This paper presents a complete study on the impact of heart volume change on the attenuation of intra-heart communications. The study was supported by COMSOL simulations and in-vitro measurements.

The topic (multi-capsule LCPs) and the paper contribution are interesting and promising, however in my opinion some improvements are needed on the paper before publishing. Herein below my detailed comments:

1-One of the most critical problems for intra-heart communications is capsule alignment. This aspect is not analysed in the paper, i recommend to add precisions/study on this point. 
2-The authors simulated the impact of the receiver resistance. However, no analytical impact is provided to explain the obtained results. I guess the behavior depends on the tissue impedance which depends on the heart model but also on the capsule size. An analysis of this point would be a good contribution in my opinion.
3-Figure 9 presents the impact of the distance between TX and RX on the attenuation. I am wondering how the simulation was carried out, was the heart size modified or was the capsule position in the heart modified? I think it is important to clarify the simulation setup
4-The measurement setup is interesting especially the dynamic modification of the blood volume in the heart. However, in my opinion, performing just measurements with a VNA is not sufficient. Actually, it would be interesting to perform transient measurements to see the impact of the attenuation during the heart cycle.
5-There are some minor typos (space between M Ohm)  with the resistance values in section 3.1.1, please correct them.

To conclude, i will finish with a long term remark, i hope that the authors will be able to validate their model with in-vivo measurements. This would constitute the best proof of their model validity.

Author Response

Response to Reviewer 1 Comments

Point 1: One of the most critical problems for intra-heart communications is capsule alignment. This aspect is not analysed in the paper, i recommend to add precisions/study on this point.

Response 1: We studied the effects of different relative angles on attenuation. The results show that when the relative angle is less than 60 °, the change of relative angle has little effect on the attenuation. Therefore, in the model and experiment, we refer to the growth direction of left ventricle and right ventricle to be consistent with the actual implantation.

Point 2: The authors simulated the impact of the receiver resistance. However, no analytical impact is provided to explain the obtained results. I guess the behavior depends on the tissue impedance which depends on the heart model but also on the capsule size. An analysis of this point would be a good contribution in my opinion.

Response 2: Thank the reviewers for pointing out this problem. An analysis of the results has been added to the article.

Point 3: Figure 9 presents the impact of the distance between TX and RX on the attenuation. I am wondering how the simulation was carried out, was the heart size modified or was the capsule position in the heart modified? I think it is important to clarify the simulation setup

Response 3: In studing the impact of the distance between TX and RX on the attenuation, we only modify the capsule position in the heart and keep the heart and chambers size still.

Point 4: The measurement setup is interesting especially the dynamic modification of the blood volume in the heart. However, in my opinion, performing just measurements with a VNA is not sufficient. Actually, it would be interesting to perform transient measurements to see the impact of the attenuation during the heart cycle.

Response 4: In the measurement setup, the sweep time is about 60ms, which is far less than the time required for a cardiac cycle (about 0.8s). At this time, the measurement using VNA is the same as the transient measurement.

Point 5: There are some minor typos (space between M Ohm)  with the resistance values in section 3.1.1, please correct them.

Response 5: Thank the reviewers for pointing out this problem. Corrections have been made in the paper.

Reviewer 2 Report

The paper is of interest and the findings may potentially be of importance in the future developments of leadless multicapsule pacemakers. I believe there are a number of changes that should be made and some questions answered. The English language is relatively good, but an English speaker/writer should go through it to improve it. I have the following questions and comments to the authors

1) Figure 1. It is mentioned that the capsules are placed inside the heart chambers. In clinical practice the LV capsule is placed in the coronary veins which are on the outside of the LV. This would create further distance between the RV and the LV including two layers of myocardium (the septum and the wall of the LV.

2) Table 1. It seems like the authors have done some wrong labeling here with ES volume being larger than the ED. This is supposed to be the other way around. Is this a labeling error or does it involve the calculations, which I assume in that case could have errors)?

3) The values for myocardial thickness seems inappropriate. It is stated that the thickness of the myocardial wall is 2mm and the septum 10 mm. This is not according to normal human values unless the authors are talking about the RV, but it seems like we are dealing with LV which is much thicker.

4) In the model description in line 186 to 196 it is stated that the muscle thickness is unchanged as the heart contracts. I am sure that the authors know that this is not correct, but they may have made this to simplify. Should be explained. Also during contraction the capsules may change angulation which could contribute to attenuation.

5) The description of the in vitro experiment is poor and difficult to understand. Also the figure is hard to figure out. The stepper motors connection to the hearts is not described and it is not clear how the fluid is pumped in and out of the heart. The description and the figure should be improved to clarify how the experimental model functions. This should include visualization of how the capsules are placed. The description is now indicating that the heart was contracting, and that the ventricle was filled from the left atrium. I cannot understand this description which I assume is incorrect. The section needs to be rewritten.

6) In the Result section only the septum is included as far as impact of the muscle is concerned. As noted earlier the path between the RV and LV capsules would normally be expected to include the LV free wall as well as the septum. Since both capsules are sitting close to the apex, the blood channel would probably be very small and the majority of the channel be muscle, since there is little blood between the two capsules. This should be discussed. The thickening of the LV muscle is substantial between systole and diastole and especially in systole there may be very little blood between the two capsules. As pointed out the communication between the RV and RA capsules, the communication may be expected to go mainly through the blood.

7) The discussion of the similarities between the simulation and the experiments should be made more detailed, mentioning the limitations of the experimental model

Author Response

Response to Reviewer 2 Comments

Point 1: Figure 1. It is mentioned that the capsules are placed inside the heart chambers. In clinical practice the LV capsule is placed in the coronary veins which are on the outside of the LV. This would create further distance between the RV and the LV including two layers of myocardium (the septum and the wall of the LV.

Response 1: Thank experts for raising this question. The implantation position of traditional pacemaker and leadless pacemaker is different. Due to the particularity of GCCIC and the good conductivity of blood, we found that placing pacemaker inside the heart chambers can effectively improve the attenuation of conduction. Therefore in our studies the capsules are wrapped in blood.

Point 2: Table 1. It seems like the authors have done some wrong labeling here with ES volume being larger than the ED. This is supposed to be the other way around. Is this a labeling error or does it involve the calculations, which I assume in that case could have errors)?

Response 2: Thank experts for pointing out this problem. In the study, we found that the change of atrial volume in cardiac cycle had little effect on attenuation. Therefore, we carried out the simulation again. In the simulation, only the ventricular volume is changed, and the atrial volume remain unchanged in cardiac cycle. The results are consistent with those before. The volume of each chamber has been corrected in Table 1.

Point 3: The values for myocardial thickness seems inappropriate. It is stated that the thickness of the myocardial wall is 2mm and the septum 10 mm. This is not according to normal human values unless the authors are talking about the RV, but it seems like we are dealing with LV which is much thicker.

Response 3: In the model, the thickness of the outer myocardium was set to 3 mm while the thickness of the interventricular septum was set to 10 mm. In this study, the septum usually refers to the septum between LV and RV.

Point 4: In the model description in line 186 to 196 it is stated that the muscle thickness is unchanged as the heart contracts. I am sure that the authors know that this is not correct, but they may have made this to simplify. Should be explained. Also during contraction the capsules may change angulation which could contribute to attenuation.

Response 4: Because attenuation fluctuation of the channel due to the change of the thickness of the interventricular septum is less than 0.5 dB, in dynamic simulation we don’t change the thickness of the septum. And we studied the effects of different relative angles on attenuation. The results show that when the relative angle is small (less than 60 °), the change of relative angle has little effect on the attenuation.

Point 5: The description of the in vitro experiment is poor and difficult to understand. Also the figure is hard to figure out. The stepper motors connection to the hearts is not described and it is not clear how the fluid is pumped in and out of the heart. The description and the figure should be improved to clarify how the experimental model functions. This should include visualization of how the capsules are placed. The description is now indicating that the heart was contracting, and that the ventricle was filled from the left atrium. I cannot understand this description which I assume is incorrect. The section needs to be rewritten.

Response 5: Thank the reviewers for pointing out this problem. This paper has been supplemented and the description has been modified.

Point 6: In the Result section only the septum is included as far as impact of the muscle is concerned. As noted earlier the path between the RV and LV capsules would normally be expected to include the LV free wall as well as the septum. Since both capsules are sitting close to the apex, the blood channel would probably be very small and the majority of the channel be muscle, since there is little blood between the two capsules. This should be discussed. The thickening of the LV muscle is substantial between systole and diastole and especially in systole there may be very little blood between the two capsules. As pointed out the communication between the RV and RA capsules, the communication may be expected to go mainly through the blood.

Response 6: As Response 4, the change of ventricular septal thickness has no obvious effect on attenuation. The change of attenuation in cardiac cycle is mainly caused by the change of distance between pacemakers.

Point 7: The discussion of the similarities between the simulation and the experiments should be made more detailed, mentioning the limitations of the experimental model

Response 7: Thank the reviewers for pointing out this problem. This paper has been supplemented.
